# Polyamine-Rich Diet Elevates Blood Spermine Levels and Inhibits Pro-Inflammatory Status: An Interventional Study

**DOI:** 10.3390/medsci9020022

**Published:** 2021-03-29

**Authors:** Kuniyasu Soda, Takeshi Uemura, Hidenori Sanayama, Kazuei Igarashi, Taro Fukui

**Affiliations:** 1Department Cardiovascular Institute for Medical Research, Saitama Medical Center, Jichi Medical University, 1-847, Amanuma, Saitama-City, Saitama 330-0834, Japan; sanayama@jichi.ac.jp (H.S.); d1423@jichi.ac.jp (T.F.); 2Amine Pharma Research Institute, Innovation Plaza at Chiba University, 1-8-15 Inohana, Chuo-ku, Chiba 260-0856, Japan; uemura@amine-pharma.com (T.U.); iga16077@faculty.chiba-u.jp (K.I.); 3Graduate School of Pharmaceutical Sciences, Chiba University, 1-8-1 Inohana, Chuo-ku, Chiba 260-8675, Japan

**Keywords:** high polyamine diet, Natto, spermine, spermidine, aging

## Abstract

The Japanese diet and the Mediterranean diet are rich in polyamines (spermidine and spermine). Increased polyamine intake elevated blood spermine levels, inhibited aging-associated pro-inflammatory status (increases in lymphocyte function-associated antigen-1 (LFA-1) on immune cells), suppressed aberrant gene methylation and extended the lifespan of mice. To test the effects of increased polyamine intake by humans, 30 healthy male volunteers were asked to eat polyamine-rich and ready-to-eat traditional Japanese food (natto) for 12 months. Natto with high polyamine content was used. Another 27 male volunteers were asked not to change their dietary pattern as a control group. The volunteers’ age of intervention and control groups ranged from 40 to 69 years (median 48.9 ± 7.9). Two subjects in the control group subsequently dropped out of the study. The estimated increases in spermidine and spermine intakes were 96.63 ± 47.70 and 22.00 ± 9.56 µmol per day in the intervention group, while no changes were observed in the control group. The mean blood spermine level in the intervention group gradually rose to 1.12 ± 0.29 times the pre-intervention level after 12 months, and were significantly higher (*p* = 0.019) than those in the control group. Blood spermidine did not increase in either group. LFA-1 on monocytes decreased gradually in the intervention group, and there was an inverse association between changes in spermine concentrations relative to spermidine and changes in LFA-1 levels. Contingency table analysis revealed that the odds ratio to decrease LFA-1 by increased polyamine intake was 3.927 (95% CI 1.116–13.715) (*p* = 0.032) when the effect of acute inflammation was excluded. The results in the study were similar to those of our animal experiments. Since methylation changes of the entire genome are associated with aging-associated pathologies and our previous studies showed that spermine-induced LFA-1 suppression was associated with the inhibition of aberrant gene methylation, the results suggest that dietary polyamine contributes to human health and longevity.

## 1. Introduction

Aging is associated with increased pro-inflammatory status, and the resultant persistent chronic inflammation is an underlying condition that is responsible for many aging-associated pathologies [1,2]. In addition, aberrant methylation changes of the entire genome are associated with aging-associated pathological changes in humans [3,4]. Increased demethylation of gene promoter enhances transcription and increased methylation diminishes this process. Therefore, demethylation of genes involved in the progression of aging-associated pathologies and methylation of genes involved in the inhibition of aging-associated pathologies promotes these pathologies. In fact, aberrant gene methylation is closely associated with both the progression of aging-associated pathologies and the lifespan of humans [5,6]. Nutritional components and macrobiotics in healthy diets have biological roles in inhibiting an aging-associated pro-inflammatory status, but the key nutritional components and dietary patterns and their effects on human health are still uncertain.

The Japanese and Mediterranean diets are rich in polyamines and associated with reduced risks of aging-associated diseases [7]. Spermidine (SPD) and spermine (SPM) are polyamines that are synthesized from arginine in all cells from microbial organisms to humans and are indispensable for cell growth, differentiation, and cellular function. In addition to the de novo synthesis, cells can take up polyamines from the surroundings. Dietary polyamines in the intestinal lumen are absorbed quickly and distributed to organs and tissues as the major source of polyamines in vivo [8,9]. Foods contain polyamines at a wide variety of concentrations, with soybeans, mushrooms, vegetables, and several fish and shellfish being rich in polyamines [10,11] and commonly eaten in Mediterranean countries and Japan [7].

Polyamines have many biological activities that protect cells and genes from harmful stimuli and counteract the pathogenesis of aging-associated pathologies [12]. We have found that polyamines, especially SPM, decrease expression of lymphocyte function-associated antigen (LFA-1) on immune cells and strongly inhibit the production of pro-inflammatory cytokines upon stimulation [13,14]. LFA-1, which consists of an alpha-L chain (CD11a) and a beta-2 chain (CD18), is a membrane protein involved in cell-cell adhesion. This adhesion activates immune cells and subsequent production of pro-inflammatory cytokines. One feature of a typical aging-associated pro-inflammatory status is increased LFA-1 expression on immune cells, which is accompanied by increased cell adhesion capacity [15,16]. The SPM concentration-dependent decrease in LFA-1 expression accompanied a decrease in the adhesion of peripheral blood mononuclear cells (PBMCs) [13].

Lifelong consumption of polyamine-rich chow with polyamine concentrations about 2 to 3 times higher than those of soybeans increased blood polyamine (mainly SPM) levels and inhibited aging-associated pathologies [17]. Mice fed high polyamine chow lived longer than other groups of mice, and aging-associated increases in LFA-1 protein levels and enhanced aberrant methylation of the entire genome were significantly suppressed in mice with a longer lifespan [18]. Our previous in vitro studies also showed that increased LFA-1 expression induced by polyamine depletion was associated with increased demethylation of the LFA-1 promoter and enhanced aberrant methylation of the entire genome. In contrast, decreased LFA-1 expression induced by SPM supplementation was associated with increased LFA-1 promoter methylation and inhibition of aberrant methylation induced by polyamine depletion [19].

This study examines whether increased polyamine intake elevates blood polyamine levels in humans and provokes the biological effects of polyamine in humans. To test the effects of high polyamine diet, natto was used for the study. Natto, a traditional Japanese food of which there have been historical descriptions for at least 400 years, is made from soybeans and has particularly high polyamine concentrations. Natto is also convenient for an interventional study because it is ready to eat with no need for cooking.

## 2. Materials and Methods

### 2.1. Study Design and Volunteers

A new form of natto using polyamine-rich soybeans and a suitable production method was developed for the study [20]. None of the soybeans or fungi were transgenic. After flavor tests, the natto contained 1880 nmol/g of spermidine and 390 nmol/g of spermine. The study was approved by the ethics committee of Saitama Medical Center (No. Rin15-35) and registered at the Center for Clinical Trials of the Japan Medical Association (JMACCT CTR; ID: JMA-IIA00233, Title: Trial to confirm biological activities in humans after long term intake of polyamine-rich food -interventional trial using newly developed fermented soybeans “Natto”-; https://dbcentre3.jmacct.med.or.jp/JMACTR/App/JMACTRS06/JMACTRS06.aspx?seqno=5483 (accessed on 29 March 2021) before the recruitment of subjects.

Healthy male volunteers aged 40 to 70 were chosen as subjects because blood polyamine levels in females change during the menstrual cycle [21]. After a verbal and written explanation of the study, volunteers were asked to join one arm of the study. Those who wanted to join the intervention group were asked to eat 45 to 90 g (1 to 2 packs) of the polyamine-rich natto every day for one year. Volunteers who joined the control group were asked not to change their dietary pattern. We asked all subjects to record all meals and snacks as images or in writing for 14 consecutive days and send records four times: before, and after 4, 8, and 12 months of the study. Blood sampling was performed at these time points.

### 2.2. Blood Tests

Blood tests were commissioned to SRL (a clinical laboratory testing company, Tokyo, Japan). High-density lipoprotein (HDL) cholesterol (mg/dL) and low-density lipoprotein (LDL) cholesterol (mg/dL) were measured by direct methods, and total cholesterol (T-Cho) (mg/dL) was measured using the Ultra Violet-End method using cholesterol dehydrogenase. High sensitivity C-reactive protein (hs-CRP) (ng/mL) was measured by a chemiluminescent enzyme immunometric assay (Latex-enhanced immunoturbidimetric assay). A lymphocyte activation test upon concanavaline-A (Con-A) stimulation was performed, and the stimulation index (SI) was calculated using radioactivity (^3^H-thymidine) uptake of stimulated blood cells divided by that of non-stimulated blood cells.

### 2.3. Flow Cytometric Analysis

We isolated PBMCs by density gradient centrifugation, using SepMate-50™ (Veritas Corp. Tokyo, Japan) and Lymphoprep™ (Axis-Shield PoC AS, Oslo, Norway). Five mL of blood was diluted by 5 mL of PBS. Diluted blood was poured down the side wall of the tube (SepMate-50) in which 15 mL of lymphoprep was filled in the lower chamber. PBMCs isolated after centrifugation (1200× *g* for 10 min) were washed twice with PBS and resuspended in the stain buffer (BD Pharmingen™) at a concentration of 1 × 107 cells/mL. FITC-conjugated anti-human CD11a antibody (BD Pharmingen™) or isotype control (20 µL per 1 × 107 cells) was added to the cell suspension. A FACScan flow cytometer (FACSVerse™) with software (BD FACSuite) was used for analysis. Expression of CD11a on 3 × 104 cells gated in the lymphocyte and monocyte light-scattered areas was analyzed. Data are expressed as mean fluorescent intensities (MFIs).

### 2.4. Determination of Polyamine Concentrations in Whole Blood

Whole blood cells were degraded by sonication and a freeze-thaw cycle, and stored at −80 °C. We determined polyamine concentrations by high-performance liquid chromatography (HPLC), after homogenization and extraction of polyamines with 0.2 N trichloroacetic acid (TCA) and centrifugation at 27,000× *g* for 15 min at 4 °C. Polyamines in 10 µL of the TCA supernatant were separated on a Toyo Soda HPLC system with a TSK gel IEX215 column (4 × 80 mm) at 50 °C. The flow rate of the buffer (0.35 M citric acid buffer [pH 5.35, 2 M NaCl, 20% methanol) was 0.35 mL/min. Polyamines were detected by fluorescence intensity after the reaction of the column effluent at 50 °C with a solution containing 0.06% o-phthalaldehyde, 0.4 M boric buffer (pH 10.4), 0.1% Brij 35, and 37 mM 2-mercaptoethanol. The flow rate of the o-phthalaldehyde solution was 0.8 mL/min, and fluorescence was measured at an excitation wavelength of 388 nm and an emission wavelength of 410 nm. The retention times for spermidine and spermine were 15 and 27 min. The concentrations are expressed as µM.

### 2.5. Statistical Analysis

Data are expressed as mean ± SD of the number (n) of samples, and some data are also shown as a percentage. Averages of two groups were compared by unpaired *t*-test for data with a normal distribution and by Mann–Whitney test when data for the two groups were unequally distributed, with *p* < 0.05 considered significant in all analyses. The odds ratio to decrease LFA-1 by increased polyamine intake was tested by 2 × 2 contingency table analysis. The relationship between two parameters was examined by linear regression analysis.

Regarding the data of subjects who dropped out the study, we also evaluated using the last observation carried forward method (LOCF) and mean imputation method (MIM) approaches.

## 3. Results

### 3.1. Subjects and Pre-Intervention Parameters

The intervention and control groups included 30 and 27 males, respectively, with ages ranging from 40 to 69 years, with a median age of 48.91 ± 7.89; 50.20 ± 7.85 in the intervention group and 47.48 ± 7.84 in the control group (*p* = 0.993, *t*-test). Analyses of the pre-intervention data showed that age was not significantly related to levels of blood spermine (SPM) with a negative *r* value (*r* = −0.150, *p* = 0.267) (SPM = 2.505 − 0.012 × age) (Figure 1a), blood spermidine (SPD) with a positive *r* value (*r* = 0.142, *p* = 0.295) (SPD = 3.419 + 0.025 × age) (Figure 1b), and the spermine to spermidine ratio (SPM/SPD) with a negative *r* value (*r* = −0.192, *p* = 0.154) (SPM/SPD = 0.660 − 0.0045 × age) (Figure 1c), while there was no statistical significance. MFIs of CD11a in the monocyte light-scattered areas (mono-CD11aMFI) correlated negative with SPM (*r* = −0.287, *p* = 0.030) (mono-CD11aMFI = 2163.00 − 100.97 × SPM) (Figure 1e) and with SPM/SPD ratio (*r* = −0.252, *p* = 0.058) (Figure 1g). Mono-CD11aMFIs had no correlation with SPD (*r* = −0.059, *p* = 0.662) (Figure 1f). While no correlation was found between mono-CD11aMFI and age (*r* = 0.080, *p* = 0.557) (Figure 1d), CD11a expression on cells in the lymphocyte light-scattering area (lymph-CD11aMFI) was positively correlated with age (*r* = 0.532, *p* < 0.001) (Figure 1h). Lymph-CD11aMFIs had no significant correlation with SPM (*r* =−0.095, *p* = 0.482) (Figure 1i), SPD (*r* = −0.081, *p* = 0.550) (Figure 1j), or SPM/SPD ratio (*r* = −0.007, *p* = 0.960) (Figure 1k).

### 3.2. Blood Tests at Each Time Point

Measured values of subjects before the intervention and 4, 8, and 12 months after intervention are shown in Table 1. The mean blood SPM concentration in the intervention group was lower than that in the control group before intervention, but increased gradually during the intervention and became higher than that of the control group at 12 months, but without significance (Table 1) (*p* = 0.958 at 4 months, *p* = 0.835 at 8 months, *p* = 0.692 at 12 months by LOCF, and *p* = 0.982 at 4 months, *p* = 0.842 at 8 months, *p* = 0.675 at 12 months by MIM). SPD did not differ significantly between the two groups in blood tests at all time points. Mono-CD11aMFI was significantly higher in the intervention group before the intervention (2040.20 ± 185.55 vs. 1891.59 ± 241.72, *p* = 0.011) and at 4 months (1851.90 ± 174.40 vs. 1716.58 ± 242.24, *p* = 0.019), while lymph-CD11aMFI did not differ. hs-CRP was high in the intervention group at 12 months, but with no significant difference from the control group. SI, HDL, LDL and T-Cho did not differ significantly between the two groups in blood tests at all time points (Table 1). Two subjects in the control group subsequently dropped out of the study.

### 3.3. Changes in Parameters during the Intervention

Changes in parameters (measured values) at 4, 8, and 12 months relative to pre-intervention values are shown in Table 2. In the intervention group, the relative mean blood SPM concentrations at 4, 8, and 12 months were higher in the intervention group than in the control group, with a significant difference at 12 months (1.12 ± 0.29 (range 0.804–2.358) vs. 0.97 ± 0.19 (0.682–1.429), *p* = 0.019). In contrast, mean blood SPD levels did not change in either group and were similar in the two groups throughout the study. Thus, the blood SPM/SPD ratio in the intervention group tended to become higher than that in the control group at 8 and 12 months, but with no significance (*p* = 0.105 at 12 months) (Table 2).

hs-CRP increased (2733.60 ± 11253.34) in the intervention group, but decreased in the control group (−510.44 ± 1589.16) after 12 months, with a significant difference in these data (*p* = 0.007). Mono-CD11aMFI decreased over time in the intervention group, but the value at each time point did not differ significantly from the corresponding value in the control group. Numerical changes of SI, HDL, LDL and T-Cho were not significant and were similar in the intervention and control groups (Table 2).

### 3.4. Relationship between Increases in Polyamine Intake and Changes in Blood Spermine Levels

Thirty subjects in the intervention group and 24 in the control group submitted meal records. Some consumed commercial natto, and the polyamine concentrations in commercial natto were estimated to be 1400 nmol/g SPD and 190 nmol/g SPM using average polyamine levels in natto.

Figure 2 shows the results evaluated using the last observation carried forward method (LOCF) approach. The estimated increases in SPD and SPM intake in the intervention group were 96.63 ± 47.70 and 22.00 ± 9.56 µmol/day, respectively, while those in the control group decreased slightly by −2.63 ± 11.60 and −0.34 ± 1.48 µmol/day, respectively, with a significant difference in the changes in polyamine intake between the two groups (*p* < 0.001).

The changes (values at 12 months after intervention minus values at pre-intervention) of polyamine (SPD plus SPM), SPD, and SPM intakes were compared with the relative changes of blood polyamine (SPD plus SPM), SPD, and SPM levels at 12 months after intervention to those of pre-intervention.

Changes in polyamine (SPD plus SPM) intake after intervention had a positive correlation (*r* = 0.335, *p* = 0.015) with blood SPM levels at 12 months relative to pre-intervention levels (Figure 2a). However, there were wide inter-individual differences in increases of blood SPM in response to increased polyamine intake. Increases in polyamine intake were not significantly correlated with changes in blood SPD (*r* = 0.215, *p* = 0.109) (Figure 2b) or the SPM/SPD ratio (*r* = 0.214, *p* = 0.110) (Figure 2c) at 12 months.

Changes in SPD intake had a positive correlation (*r* = 0.353, *p* = 0.007) with changes in blood SPM (Figure 2d). Increases in SPD intake had a low positive *r* value with changes of blood SPD (*r* = 0.215, *p* = 0.109) (Figure 2h) and the relative SPM/SPD ratio (*r* = 0.209, *p* = 0.119) (Figure 2i), while there was no statistical significance.

Changes in SPM intake after the intervention had a positive correlation (*r* = 0.364, *p* = 0.005) with changes of blood SPM at 12 months after intervention (Figure 2g). Increases in SPM intake had a low positive *r* value with both the relative concentrations of blood SPD (*r* = 0.204, *p* = 0.128) (Figure 2e) and the relative SPM/SPD ratio (*r* = 0.234, *p* = 0.075) (Figure 2f), while there was no statistical significance.

The results were the same when raw data of 30 volunteers in the intervention group and 24 in the control group were analyzed. In addition, when we evaluated using the mean imputation method (MIM) approach, the results were the same.

### 3.5. Changes in Parameters of Subjects Without Acute Inflammation and Evaluation of the Effects of SPM on LFA-1 Expression in Monocytes

Inflammation significantly affects LFA-1 expression (Appendix A) and hs-CRP is a marker of inflammation, with an acute increase in hs-CRP, indicating acute inflammation. Therefore, an analysis was performed after the exclusion of subjects in whom hs-CRP at 4, 8, 12 months changed (increased or decreased) by more than 3000 ng/mL from the pre-intervention value (Table 3). In this analysis, the mean value of the relative blood SPM levels at 12 months compared to pre-intervention increased significantly by 1.08 ± 0.18 times (*p* = 0.019) in the intervention group, while those in the control group did not change (0.98 ± 0.20 times). The mean mono-CD11aMFI in the intervention group tended to decrease with time, and the change at 12 months was significantly lower than that in the control group (−247.83 ± 148.76 vs. −85.04 ± 257.39, *p* = 0.019).

Relative blood SPM levels at 12 months compared to pre-intervention showed a negative relationship with changes in mono-CD11a MFIs from pre-intervention to 12 months after intervention in all subjects, but without significance (*p* = 0.114). The same lot of CD11a antibody for flow cytometry could not be used throughout the intervention due to the limited expiration date. To correct the difference of antibody titer, individual pre-intervention mono-CD11aMFIs were adjusted (estimated values) by the difference (−164.16) in the mean MFI between pre-intervention and 12 months, and the individual actual values of mono-CD11aMFI at 12 months were compared to the estimated values in subjects with a hs-CRP change of <3000 ng/mL. In the intervention group, the actual values were higher than the estimated values in 5 subjects and lower than the estimated values in 18 subjects. In the control group, 12 subjects had higher and 11 had lower actual values compared to estimated values. Contingency table analysis revealed that the odds ratio to decrease mono-CD11aMFI by increased polyamine intake was 3.927 (95% CI 1.116–13.715) (*p* = 0.032 by Pearson uncorrected test).

### 3.6. Evaluation of the Effects of Changes in Polyamine Concentrations on LFA-1 Expression in the Monocyte Area

We further evaluated the effects of changes in polyamine concentrations on LFA-1 expression in the monocyte area (mono-CD11aMFIs). Numerical changes at each time point (4, 8, and 12 months after intervention) from pre-intervention values of polyamines (SPD & SPM) and CD11aMFI using LOCF and MIM (in parentheses) approaches are shown in Table 4.

The change of mean blood SPM concentration after 4 (*p* = 0.063 by LOCF, *p* = 0.069 by MIM) and 8 months (*p* = 0.116 by LOCF, *p* = 0.129 by MIM) in the intervention group was greater than that of the control group, and a significant difference (*p* = 0.041 by LOCF, *p* = 0.042 by MIM) was observed at 12 months. The numerical change of blood SPD concentration at 4, 8, and 12 months from that of pre-intervention in the intervention group was similar to that of the control group throughout the study. The mean blood SPM/SPD ratio of 4, 8, and 12 months relative to that of pre-intervention value was slightly higher in the intervention group than in the control group, while no significant difference was observed.

Mean difference in mono-CD11aMFIs between before and after 4 months after intervention in the intervention group was similar to that of the control group (*p* = 0.521 by LOCF, *p* = 0.647 by MIM). However, the values in the intervention group decreased throughout the trial period, and the difference between pre-intervention and 8 and 12 months after the start of the study in the intervention group became significantly different from those in the control group at 8 and 12 months after intervention (*p* = 0.020 by LOCF, *p* = 0.026 by MIM at 8 months) (*p* = 0.006 by LOCF, *p* = 0.010 by MIM at 12 months).

The relationship between two parameters was examined by linear regression analysis using the LOCF approach. The changes in blood SPM concentrations after 12 months of intervention (values of 12 months minus those of pre-intervention values) had a negative *r* value (*r* = −0.134, *p* = 0.322) with the changes in mono-CD11aMFIs (12 months–pre-intervention) (Figure 3a). The changes in blood SPD concentrations had no correlation (*r* = −0.027, *p* = 0.843) with the changes in mono-CD11aMFIs (Figure 3b). The values of blood SPM/SPD ratio at 12 months relative to pre-intervention had a negative *r* value (*r* = −0.250, *p* = 0.061) with changes in mono-CD11aMFIs (Figure 3c). Our previous studies in which the effects of SPM on LFA-1 expressions were examined showed that SPM supplementation-induced decreases in LFA-1 expressions of cultured cells and human peripheral blood mononuclear cells were observed very gradually [13]. Therefore, changes of blood polyamine concentrations and SPM/SPD ratio at 8 months were compared with the changes of LFA-1 after 12 months of intervention. The changes in blood SPM concentrations after 8 months of intervention (changes of values at 8 months from those of pre-intervention) had a negative *r* value (*r* = −0.176, *p* = 0.191) with the changes in mono-CD11aMFIs (12 months–pre-intervention). The changes in the values of relative blood SPM/SPD ratio at 8 months to pre-intervention had a negative correlation (*r* = −0.283, *p* = 0.033) with mono-CD11aMFIs after 12 months of intervention (Figure 3d)

## 4. Discussion

The activities of enzymes involved in polyamine synthesis decrease with age. However, the aging-associated declines in polyamine concentrations were observed only during early life. Nishimura et al. found that polyamine concentrations in various tissues and organs were significantly lower in 10- and 26-week-old mice than in 3-week old mice, but no differences in SPD and SPM concentrations were observed between 10- and 26-week-old mice, except in the skin [22]. Similarly, as observed in this study, no age-associated declines have been observed in blood polyamine concentrations and in urine polyamine excretion in adult humans [13,23,24,25]. A large inter-individual differences in blood SPD and SPM concentrations observed in the study have also been reported in previous studies [13,23], although exact biological mechanisms underlying the inter-individual differences are unknown.

The results of a previous study indicated that cells in which SPM suppressed LFA-1 expression differed from those with increased LFA-1 with aging [13]. This study showed that aging-associated increases and SPM concentration-dependent decrease in LFA-1 protein levels occur in different cell populations. In this study, we isolated PBMCs immediately after blood sampling, which reduced the time for PBMCs to contact the plastic wall of the blood collection tube. Cell-to-wall contact stimulates PBMCs, and however weak this stimulation is, it enhances LFA-1 expression and adhesion capacity via the production of a small amount of pro-inflammatory cytokines. Such interactions augment the effects of SPM on LFA-1 expression on all immune cells because polyamine (especially SPM) inhibit pro-inflammatory cytokine synthesis [14]. Rapid isolation enabled us to reveal that aging-associated increases in LFA-1 protein levels were observed in the lymphocyte area, while the effect of SPM on LFA-1 was obvious in the monocyte area.

The weakness of the study is that the study was not conducted in an intervention assignment or by randomization. Natto is a traditional Japanese food with a peculiar odor. Like other food preferences of this kind, preferences are roughly different from person to person. Therefore, we considered that one-year natto intake is not possible when volunteers who do not like natto were assigned in the study group, and there must be a large number of dropout cases. As a result, no one in the study group dropped out, and the study first revealed the effects of long-term increased polyamine intake in humans. Estimated average intakes of SPD and SPM from foods in the Japanese population are 36 and 74 µmol/day, respectively [10]. In this study, increases in SPD and SPM intakes from polyamine-rich natto were comparable to the daily SPD and SPM intakes by the average Japanese person. Natto contains 4 to 5 times more SPD than SPM, but blood SPD levels did not increase. The exact mechanisms through which increased polyamine intake predominantly elevates blood SPM levels are unclear, but our animal experiments showed similar results: polyamine-rich chow containing 1540 nmol/g of SPD and 374 nmol/g of SPMe also predominantly elevated blood SPM in mice [18,26]. Blood SPD increased in a few mice, but without statistical significance.

The relationship between increases in polyamine intake and those in blood SPM levels seems not to be simply additive. Differences in increases of blood SPM in response to increased polyamine intake were also observed in animal experiments, despite all mice being born and bred in the same environment and fed the same chow [17,18]. Extracellular polyamine supply has a significant effect on intracellular polyamine concentrations, a phenomenon that is typically seen in patients with cancer. Polyamine biosynthesis is up-regulated in cancer cells, and therefore SPD concentrations, which reflect the activity of cell growth, are increased in the blood and urine in patients with cancer. These facts indicate that SPM supply is increased by increased polyamine intake. There are two sources of polyamines in the intestinal lumen: those in ingested foods and those synthesized by microbes. Recent studies indicate the importance of the composition of the intestinal microbiota for synthesis of intestinal polyamines [27]. Matsumoto et al. reported that probiotics administration increased SPM, but not SPD, concentrations in feces in humans and animals, although probiotics and intestinal microbiota by themselves cannot synthesize SPM [28,29]. The different responses to increased polyamine intake may reflect differences in the intestinal environment, and especially in the composition of the intestinal microbiota.

Both SPD and SPM have superficially similar biological activities. However, experiments using PBMCs and cultured immune cells have shown that SPM has greater potency for decreasing pro-inflammatory cytokine synthesis and LFA-1 expression, compared to SPD. SPM has potent effects when intracellular concentrations are increased about 1.2 to 1.3 times, while SPD only has similar activities at intracellular concentration increases of 2 to 3 times [13]. In this study, increased polyamine intake increased mean SPM levels by more than 1.1 times, but did not increase SPD, which suggests that biological effects induced by increased polyamine intake are mainly ascribable to SPM.

LFA-1 protein level is not only regulated by ITGAL (LFA-1 promoter) methylation but also controlled by an intracellular signaling pathway [30], and inflammation typically increase LFA-1 levels. Therefore, volunteers who had acute inflammatory conditions at the time of blood sampling were eliminated based on exponential changes of hs-CRP that reflect acute inflammation. This elimination resulted in the effects of SPM on LFA-1 expressions becoming significant in subjects. Furthermore, the inverse correlations between the changes in LFA-1 expression and the changes in SPM concentrations or the changes in SPM/SPD ratio were further confirmed, even when the analyses were performed ignoring the effects of inflammation (Figure 3). These findings are similar to those in our animal experiments, in which increased blood SPM levels by increased polyamine intake accompanied by inhibition of aging-associated increases in LFA-1 protein levels and inhibition of enhanced aberrant methylation of the entire genome [17].

Global alterations in DNA methylation status associated with age have been documented in many vertebrate tissues [31,32], and such changes are considered a major cause of age-associated chronic diseases and fragility [33,34,35]. Aging-associated or polyamine depletion-induced enhancement of aberrant methylation of the entire genome is associated with increased demethylation of the LFA-1 promoter and increased LFA-1 protein levels. In contrast, decreased LFA-1 expression induced by SPM was associated with increased LFA-1 promoter methylation and inhibition of aberrant methylation [19]. These indicate that the suppression of LFA-1 by increased polyamine intake is associated with suppression of aberrant methylation of the entire genome.

## 5. Conclusions

Increased polyamine (spermidine and spermine) intake by natto elevated blood spermine levels and inhibited pro-inflammatory status.

## Figures and Tables

**Figure 1 medsci-09-00022-f001:**
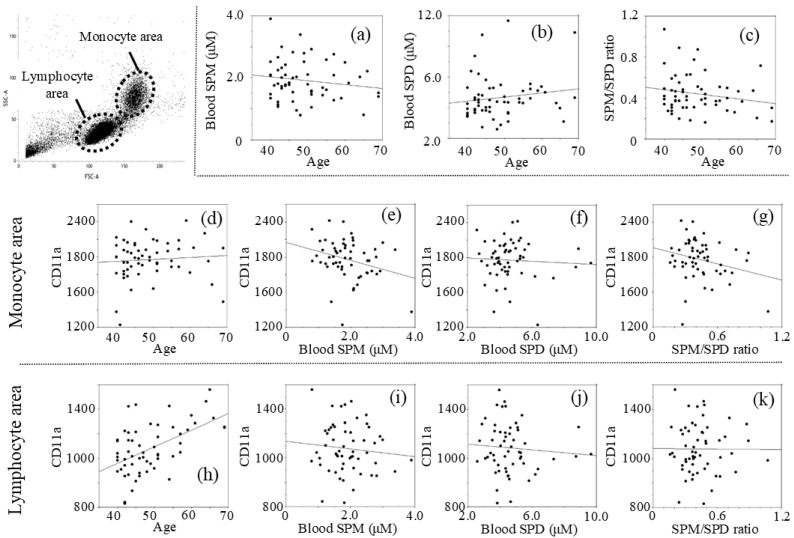
Relationship between two measurements of pre-intervention: (**a**) Age vs. SPM; (**b**) Age vs. SPD; (**c**) Age vs. SPM/SPD; (**d**) Age vs. mono-CD11aMFI; (**e**) SPM vs. mono-CD11aMFI; (**f**) SPD vs. mono-CD11aMFI; (**g**) SPM/SPD ratio vs. mono-CD11aMFI; (**h**) Age vs. lymph-CD11aMFI; (**i**) SPM vs. lymph-CD11aMFI; (**j**) SPD vs. lymph-CD11aMFI; (**k**) SPM/SPD ratio vs. lymph-CD11aMFI. Concentrations of polyamines were measured by HPLC and expression of CD11a (LFA-1) was determined using a FACScan flow cytometer (FACSVerse^TM^) with analysis software (BD FACSuite™). SPM; spermine concentrations, SPD; spermidine concentrations, mono-CD11a; LFA-1 (CD11a) expression on monocytes, lymph-CD11a; LFA-1 (CD11a) expression on lymphocytes, MFI; mean fluorescent intensities.

**Figure 2 medsci-09-00022-f002:**
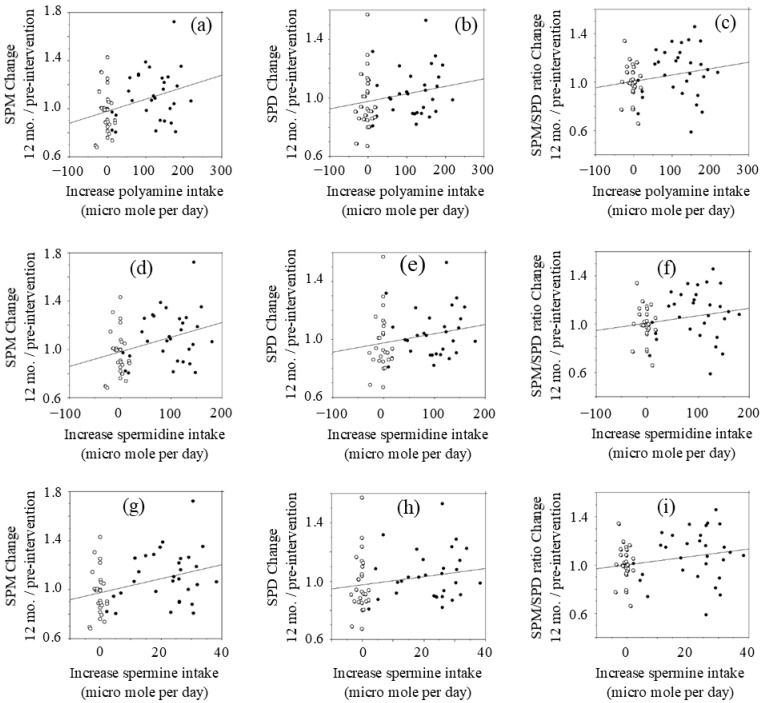
Relationship between changes in polyamine intake and changes in blood polyamine (LOCF approach).: (**a**) blood SPM changes vs. changes in polyamine intake; (**b**) blood SPD changes vs. changes in polyamine intake; (**c**) blood SPM/SPD changes vs. changes in polyamine intake; (**d**) blood SPM changes vs. changes in SPD intake; (**e**) blood SPD changes vs. changes in SPD intake; (**f**) blood SPM/SPD changes vs. changes in SPD intake; (**g**) blood SPM changes vs. changes in SPM intake; (**h**) blood SPD changes vs. changes in SPM intake; (**i**) blood SPM/SPD changes vs. changes in SPM intake. Concentrations of polyamines were measured by HPLC. SPM; spermine, SPD; spermidine. Open circle; control, Closed circle; natto group

**Figure 3 medsci-09-00022-f003:**
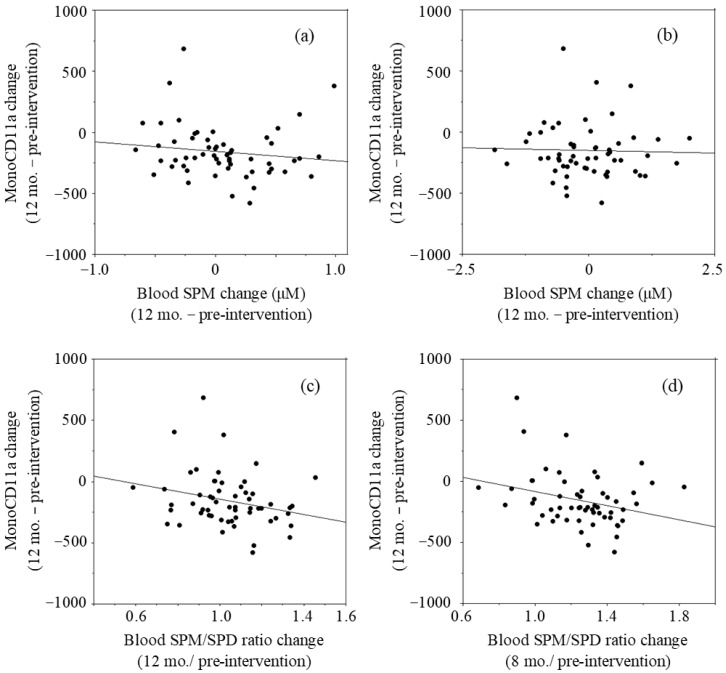
Relationship between changes in blood polyamine concentrations and changes in LFA-1 levels in monocyte area (mono-CD11aMFI).: (**a**) Changes in blood SPM concentrations at 12 months vs. changes in mono-CD11aMFI at 12 months; (**b**) Changes in blood SPD concentrations at 8 months vs. changes in mono-CD11aMFIs at months; (**c**) Changes in the blood SPM/SPD ratios at 12 months vs. changes in mono-CD11aMFIs at 12 months; (**d**) Changes in the blood SPM/SPD ratios at 8 months vs. changes in mono-CD11aMFIs at 12 months. Mono-CD11aMFI; Mean fluorescent intensities (MFI) of CD11a expressions (LFA-1) on cells in monocyte area, SPM; Spermine, SPD; Spermidine, SPM/SPD: Relative content of SPM to SPD.

**Table 1 medsci-09-00022-t001:** Measured values at each time point.

	Pre-Intervention		4 months		8 months		12 months	
Group	Intervention	Control	Intervention	Control	Intervention	Control	Intervention	Control
n	30	27	30	26	30	25	30	25
SPM	1.86 ± 0.60	1.99 ± 0.68	2.09 ± 0.88	2.13 ± 0.90	2.09 ± 0.72	2.04 ± 0.83	2.02 ± 0.67	1.93 ± 0.86
SPD	4.71 ± 1.40	4.57 ± 1.41	4.92 ± 1.65	4.50 ± 1.88	4.19 ± 1.28	3.86 ± 0.99	4.81 ± 1.18	4.44 ± 1.30
SPM/SPD	0.42 ± 0.18	0.46 ± 0.19	0.46 ± 0.23	0.50 ± 0.21	0.53 ± 0.23	0.55 ± 0.25	0.45 ± 0.20	0.46 ± 0.21
hs-CRP	1207.07 ± 4064.23	1316.26 ± 1494.05	1849.63 ± 3705.88	1561.89 ± 3924.60	742.63 ± 1369.05	818.76 ± 1098.17	3490.67 ± 10250.95	895.48 ± 1045.75
Lymph-CD11a MFI	1082.87 ± 230.87	1074.85 ± 168.84	1102.90 ± 200.65	1090.04 ± 173.25	1051.80 ± 201.22	1015.92 ± 170.99	994.17 ± 176.34	999.36 ± 143.28
Mono-CD11a MFI	2040.20 ± 185.55 *^1^	1891.59 ± 241.72	1851.90 ± 174.40 *^2^	1716.58 ± 242.24	1882.83 ± 200.45	1852.64 ± 171.63	1808.47 ± 176.95	1805.40 ± 192.56
SI	232.65 ± 104.66	213.94 ± 92.85	234.86 ± 64.44	208.65 ± 67.57	146.74 ± 49.78	159.05 ± 55.41	206.29 ± 84.99	193.29 ± 85.22
HDL	62.53 ± 15.15	65.26 ± 20.97	61.33 ± 16.16	58.35 ± 16.53	60.87 ± 17.60	58.48 ± 17.36	65.13 ± 20.93	63.16 ± 16.48
LDL	128.47 ± 39.81	121.37 ± 35.03	123.97 ± 31.79	122.62 ± 33.04	117.43 ± 27.99	118.88 ± 28.50	133.10 ± 37.10	130.6 ± 36.68
T-Cho	212.07 ± 43.17	207.89 ± 41.26	205.47 ± 33.50	200.96 ± 34.56	203.37 ± 34.49	195.80 ± 31.47	219.77 ± 38.69	216.52 ± 40.15

Data are presented as mean ± SD. n indicates the number of volunteers at each time point. The unit of each measurement is given in the Method section. Lymph-CD11aMFI; mean ± SD of CD11a MFI in lymphocyte light-scattered area, mono-CD11aMFI; mean ± SD of CD11a MFI in monocyte light-scattered area, SPM; spermine concentrations, SPD; spermidine concentrations, SPM/SPD; ratio of spermine/spermidine, hs-CRP; high sensitivity C-reactive protein, lymph-CD11a; LFA-1 (CD11a) expression on lymphocytes, mono-CD11a; LFA-1 (CD11a) expression on monocytes, MFI; mean fluorescent intensities, SI; stimulation index, HDL; high-density lipoprotein, LDL; low-density lipoprotein, T-chol; total cholesterol. *^1^
*p* = 0.011 by unpaired *t* test, *^2^
*p* = 0.019 by unpaired *t* test.

**Table 2 medsci-09-00022-t002:** Changes in parameters (measured values).

	Pre−Intervention		4 months		8 months		12 months	
Group	Intervention	Control	Intervention	Control	Intervention	Control	Intervention	Control
n	30	27	30	26	30	25	30	25
Age	50.20 ± 7.85	47.48 ± 7.84	50.20 ± 7.85	47.35 ± 7.96	50.20 ± 7.85	47.64 ± 7.98	50.20 ± 7.85	47.64 ± 7.98
SPM change rate	1	1	1.13 ± 0.32	1.07 ± 0.36	1.16 ± 0.32	1.04 ± 0.22	1.12 ± 0.29 *^1^	0.97 ± 0.19
SPD change rate	1	1	1.04 ± 0.24	0.98 ± 0.28	0.90 ± 0.18	0.85 ± 0.13	1.04 ± 0.17	0.97 ± 0.20
SPM/SPD change rate	1	1	1.09 ± 0.25	1.09 ± 0.23	1.31 ± 0.26	1.23 ± 0.21	1.09 ± 0.24	1.01 ± 0.13
hs-CRP	0	0	642.57 ± 5570.59	203.23 ± 4405.73	−464.43 ± 4318.86	−587.16 ± 1946.86	2733.60 ± 11253.34 *^2^	−510.44 ± 1589.16
Lymph- CD11aMFI	0	0	20.03 ± 101.61	13.23 ± 105.50	−31.07 ± 137.57	−62.36 ± 140.72	−88.7 ± 121.66	−78.92 ± 100.38
Mono- CD11aMFI	0	0	−188.30 ± 201.57	−169.12 ± 221.79	−157.37 ± 186.48	−35.84 ± 213.36	−231.73 ± 166.33	−83.08 ± 249.52
SI	0	0	2.20 ± 92.58	3.17 ± 89.32	−85.91 ± 115.63	−47.31 ± 71.28	−24.110 ± 134.46	−13.08 ± 78.66
HDL	0	0	−1.20 ± 7.02	−4.77 ± 6.37	−1.67 ± 7.81	−4.56 ± 8.10	2.60 ± 11.51	0.12 ± 7.89
LDL	0	0	−4.50 ± 26.89	−0.81 ± 18.97	−11.03 ± 26.45	−5.36 ± 17.12	4.63 ± 29.21	6.36 ± 20.55
T-Cho	0	0	−6.60 ± 28.08	−7.04 ± 27.79	−8.70 ± 30.30	−12.88 ± 24.34	7.70 ± 34.17	7.84 ± 26.62

Data are presented as mean ± SD. n indicates the number of volunteers at each time point. Numerical changes at each point from pre-intervention values are expressed as mean ± SD. SPD, SPM, and SPM/SPD at each time point are expressed as values relative to corresponding pre-intervention values (value at each time point/pre-intervention value). The unit of each measurement is given in the Method section., SPM; spermine concentrations, SPD; spermidine concentrations, SPM/SPD; ratio of spermine/spermidine, hs-CRP; high sensitivity C-reactive protein, lymph-CD11a; LFA-1 (CD11a) expression on lymphocytes, mono-CD11a; LFA-1 (CD11a) expression on monocytes, MFI; mean fluorescent intensities, SI; stimulation index, HDL; high-density lipoprotein, LDL; low-density lipoprotein, T-chol; total cholesterol. *^1^
*p* = 0.019 Mann-Whitney, *^2^
*p* = 0.007 Mann Whitney.

**Table 3 medsci-09-00022-t003:** Changes in parameter in subjects with no inflammation (measured values).

	Pre−Intervention		4 months		8 months		12 months	
Group	Intervention	Control	Intervention	Control	Intervention	Control	Intervention	Control
*n*	30	27	24	22	27	20	23	23
age	50.20 ± 7.85	47.48 ± 7.84	49.75 ± 7.79	46.82 ± 7.15	50.19 ± 8.09	48.85 ± 8.48	49.65 ± 7.67	47.96 ± 8.25
SPM change rate	1	1	1.10 ± 0.33	1.04 ± 0.33	1.19 ± 0.33	1.07 ± 0.22	1.08 ± 0.18 *^1^	0.98 ± 0.20
SPD change rate	1	1	1.05 ± 0.25	0.95 ± 0.22	0.90 ± 0.18	0.85 ± 0.13	1.04 ± 0.17	0.96 ± 0.20
SPM/SPD change rate	1	1	1.05 ± 0.21	1.08 ± 0.16	1.33 ± 0.26	1.28 ± 0.20	1.05 ± 0.19	1.02 ± 0.13
hs-CRP	0	0	−500 ± 624.63	−109.68 ± 885.95	−46.70 ± 588.73	−160.85 ± 556.02	−32.87 ± 482.02	−139.35 ± 879.41
Lymph- CD11aMFI	0	0	7.46 ± 104.56	17.46 ± 103.27	−38.52 ± 141.15	−65.90 ± 133.64	−99.00 ± 129.77	−70.83 ± 96.42
Mono- CD11aMFI	0	0	−209.08 ± 214.63	−177.41 ± 222.79	−156.63 ± 196.33	−47.10 ± 226.90	−247.83 ± 148.76 *^2^	−85.04 ± 257.39

Data are presented as mean ± SD. n indicates the number of volunteers at each time point. Numerical changes at each point from pre-intervention values are expressed as mean ± SD. SPM, SPD, and SPM/SPD at each time point are expressed as values relative to corresponding pre-intervention values (value at each time point/pre-intervention value). The unit of each measurement is given in the Method section. SPM; spermine concentrations, SPD; spermidine concentrations, SPM/SPD; ratio of spermine/spermidine, hs-CRP; high sensitivity C-reactive protein, lymph-CD11a; LFA-1 (CD11a) expression on lymphocytes, mono-CD11a; LFA-1 (CD11a) expression on monocytes, MFI; mean fluorescent intensities. *^1^
*p* = 0.019 Mann-Whitney, *^2^ 0.007 Mann-Whitney.

**Table 4 medsci-09-00022-t004:** Changes in polyamine concentrations and CD11a expressions in monocyte area by LOCF and MIM approaches (MIM in parentheses).

	4 months—Pre-Intervention		8 months—Pre-Intervention		12 months—Pre-Intervention	
Group	Intervention	Control	Intervention	Control	Intervention	Control
SPM (MIM approach)	0.120 ± 0.257	0.0005 ± 0.220 (−0.006 ± 0.258)	0.228 ± 0.428	0.055 ± 0.381 (0.057 ± 0.403)	0.161 ± 0.373 *^1^	−0.049 ± 0.381 (−0.052 ± 0.403)
SPD (MIM approach)	−0.182 ± 0.583	−0.268 ± 0.510 (−0.200 ± 0.602)	−0.520 ± 0.904	−0.734 ± 0.792 (−0.697 ± 0.839)	−0.093 ± 0.756	0.195 ± 0.795 (−0.116 ± 0.869)
SPM/SPD ratio (MIM approach)	1.133 ± 0.164	1.074 ± 0.137 (1.060 ± 0.164)	1.290 ± 0.225	1.214 ± 0.209 (1.210 ± 0.224)	1.078 ± 0.207	1.005 ± 0.129 (0.990 ± 0.153)
Mono−CD11aMFI (MIM approach)	−188.30 ± 201.57	−152.17 ± 220.65 (−162.70 ± 218.61)	−157.37 ± 205.24 *^1^	−33.02 ± 205.24 (−37.73 ± 207.55)	−231.73 ± 166.33 *^2^	−76.59 ± 240.87 (−86.07 ± 240.06)

Data are presented as mean ± SD. Numerical changes of values at each point from pre-intervention values were expressed as mean ± SD. SPM/SPD at each time point were expressed as values relative to corresponding values of pre-intervention (values at each time point/values of pre-intervention). SPM; spermine concentrations, SPD; spermidine concentrations, SPM/SPD; ratio of spermine/spermidine, hs-CRP; high sensitivity C-reactive protein, mono-CD11a; LFA-1 (CD11a) expression on monocytes, MFI; mean fluorescent intensities. *^1^
*p* < 0.05, *^2^
*p* < 0.01 by unpaired *t* test.

## Data Availability

The data that support the findings of this study are available from the corresponding author (Kuniyasu Soda), upon reasonable request.

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
