# Peer review of "Polyamine-Rich Diet Elevates Blood Spermine Levels and Inhibits Pro-Inflammatory Status: An Interventional Study"

_medsci, 2021, doi:10.3390/medsci9020022_

Round 1
Reviewer 1 Report
Soda et al. present data from a clinical trial studying the effects of a 1-year natto supplementation, with increased polyamine content, on blood polyamine levels, some inflammation markers and the HDL/LDL system. While the study is of great interest for the scientific and public readership, there are several questions about the study design and the analyses of the data.
Major
- I think the title is misleading and I would suggest changing it to something like „Intake of natto with increased polyamine content elevates…”
- The main conclusion of title and abstract (“The intervention increases spermine levels.”) is based on repeated t-tests and/or mann-whitney-u tests separately at each timepoint. I am not convinced that this is the best choice of analysis and would recommend consulting a dedicated statistician about this point.
- The manuscript lacks and completely ignores significant references to work by other groups with spermidine and polyamines in inflammation research, model organisms, aging research, autophagy induction and clinical trials. The authors must extend the inclusion of relevant work and lengthen their discussion to relevant preclinical and clinical studies, as well as recent reviews which discuss the field.
- The information on increased polyamine intake in the abstract does not match the information from the main text or figures. Please clarify which information is the right one. For instance, compare Line 233 with Figure 2d-i. Also if the lowest intake per day is 1 pack of natto (1880nmol/g, 45g package size), then the lower limit should be roughly 84.6µmol spermidine. Still, the average increase is stated as 22µmol/day and this also does not fit with the mentioned figures.
- The last two sentences of the abstract are highly speculative and overstated and should be omitted. The authors cannot talk about aging-associated pro-inflammatory status, when they did not include young controls.
- The authors should clarify for instance for Figure 2, if the correlations were calculated across both control and intervention groups. At least the number of data points suggests so. If this is the case, then – in my understanding – the correlations cannot be presented like this and should be evaluated only within the groups. Also, the groups should be indicated (by colour or symbol). Else, there is a high risk of creating spurious correlation artifacts. Hence, the correlations must be recalculated and additionally presented at least for the intervention group alone.
- Why did the authors not measure standard inflammatory blood markers like IFNs, TNF-alpha and so on?
- Is the exact nutritional composition of the natto used in the study known? If so, this should be mentioned here, as there are likely other bioactive compounds present.
- The suggested daily additional intake of spermidine via the natto seems very high (>12mg for 1 pack, >24mg for 2 packs). This is way beyond what other currently running clinical trials use as spermidine supplementation. Has there been any kind of safety evaluation for this amount of daily polyamine intake? This needs to be commented on, especially regarding the cancer relevance. The authors need to justify and comment on the dosage selection. Also, were there any adverse events recorded? Is this information available anywhere?
- Did the authors perform power analyses before recruiting? Please include this information.
- Is there an estimation of how much spermidine and spermine the diet of the study cohort contained over the study and what percentage the natto-supplementation increased over baseline intake levels? Should be possible to estimate based on the food records. This would be a crucial information for such studies.
- Blood levels of polyamines apparently do not correlate well with the increased polyamine intake. Why do the authors continue correlating blood polyamine changes with other parameters? Why don’t they concentrate on the intake data?
Minor
- Please explain “reinforced” natto, what does it exactly mean technically?
- The link to the trial registration does not exist/work.
- Title of registered study does not match with the one in the Materials and Methods section. Please comment on this and clarify.
- In the cancer field there is some concern of polyamines enhancing tumor growth. The authors should address this point in the discussion. This is particularly important as the dosage which was used is extremely high.
- Please include a cohort description table.
- Table 1: Are the indicated p-values the only significant ones?
- Tables 1 and 2 basically show the same data, should be combined.
- There is a large discrepancy of literature on the age-dependency of polyamine concentration in murine tissue and human samples. The authors concentrate on literature supporting their own data but should also include other reports. Especially, the data should be discussed in view of Pekar et al. and Pucciarelli et al.
- The discussion should include a dedicated and extended limitations part.
- Are there records of study compliance? E.g. how often did participants in the intervention group miss the natto intake goal?
- Line 48: “Nutritional components and macrobiotic…” A word seems to be missing after macrobiotic.
- Line 52: Please provide information about polyamine content estimations for both diets.
- Lines 71-72: Incomplete sentence, verb is missing.
- At the first mention of the inflammation parameters their relevance should be briefly introduced.
- Table legends should include explanations of the abbreviations.
- I recommend indicating r and p-values in the correlation plots.
- Lines 213-214: Unit missing for hs-CRP.
- Table 4: What is the unit of the numbers? How does this table relate to the data shown in Table 2?
- Figure 3 seems to cherry-pick time points, why not systematically analyzing the correlations across all time points?
- Another point that needs to be discussed in a limitations section is that most of the significant observations have rather high p-values, while no corrections for multiple comparisons are applied. Thus, the conclusions should be considered very preliminary and with caution. Given this, it needs to be made very clear that this study was of explorative nature.
- Discussion Line 352 should be referenced.
- Line 352: Decide to use plural or singular.
- The manuscript should be checked by a native English-speaking person.
- The molecular mechanisms of polyamines are wide and somewhat elusive, especially for external supplementation. The authors did not address some key molecular aspects of external polyamine supply, e.g. hypusination of eIF5A or autophagy induction, which could at least indirectly by examined in the isolated PBMCs. If time allows, this would increase the strength of the study but if it’s technically not possible, the variety of potential molecular mechanisms should at least be mentioned in the discussion. Also, if blood cells are available, the authors could determine polyamine levels in those samples, as they might respond differently than whole blood to the increased polyamine intake.
- The authors should show unstained controls, at least examples for e.g. Fig 1 and so on.
- Line 229-230: estimated natto polyamine levels should be referenced.
- The data on HDL/LDL should also be mentioned in the abstract.
Author Response
- I think the title is misleading and I would suggest changing it to something like „Intake of natto with increased polyamine content elevates…”
Thank you very much. We change the title of our manuscript referring to your opinions. (marked by broad yellow line)
- The main conclusion of title and abstract (“The intervention increases spermine levels.”) is based on repeated t-tests and/or mann-whitney-u tests separately at each timepoint. I am not convinced that this is the best choice of analysis and would recommend consulting a dedicated statistician about this point.
We asked the doctor who is expertise in this field, and we examined t-test and Mann Whitney tests separately at each time point. No p-value indicated when there is no significance or difference.
- The manuscript lacks and completely ignores significant references to work by other groups with spermidine and polyamines in inflammation research, model organisms, aging research, autophagy induction and clinical trials. The authors must extend the inclusion of relevant work and lengthen their discussion to relevant preclinical and clinical studies, as well as recent reviews which discuss the field.
We do not ignore works by other groups. When we refer many previous papers in which spermidine concentrations in tissues, organs, blood, urine, and semen were measured, the effects of spermine (as you know, spermidine has little ability to suppress inflammation) on inflammation, and the experimental results of model organisms and aging research, there must be too many citations. This article is not a review article, therefore, we cited our review articles in which we referred many papers reported not only by us but by other investigators.
Aging-associated decrease in polyamine concentrations described in the title or the abstract only indicate that the decrease is observed only during developmental and growth period. When you read the body of many previous papers concerning on the polyamine concentrations in tissues, organs, blood, and urine, you can find that majority of the previous paper clearly show that spermidine does not change with aging in adult. Moreover, in some reports, it was shown that spermidine concentrations in prostate and semen in humans and in pancreas in animals increase (not increase) with aging. When you read previous papers, you realize that the majority of papers clearly showed no age-dependent decrease in spermidine concentration in adult. The one problem we concern is that when serum polyamine concentrations is measured by HPLC, it sometimes hard to detect peak of polyamine, especially spermine. In some case with lower polyamine concentrations, spermine peak is detected like a shaking of the base line of HPLC. We consider it is very difficult to determine accurate polyamine concentrations using such unclear peak. As you know, serum contains only 1% of blood polyamine. Majority of the polyamine is contained in blood cells. These facts indicate that when hemolysis, no matter how slight, occurred in the blood sample, serum levels of polyamine were affected significantly.
The other curious thing is that many food components, such as so-called anti-oxidant substances and anti-oxidant vitamins failed to extend life span of mammals in spite of the fact they are absorbed in the body and they significantly activate autophagy.
In the near future, I will publish a review article in which recent works on spermidine and polyamines in inflammation research, model organisms, aging research, autophagy induction and clinical trials.
- The information on increased polyamine intake in the abstract does not match the information from the main text or figures. Please clarify which information is the right one. For instance, compare Line 233 with Figure 2d-i. Also if the lowest intake per day is 1 pack of natto (1880nmol/g, 45g package size), then the lower limit should be roughly 84.6µmol spermidine. Still, the average increase is stated as 22µmol/day and this also does not fit with the mentioned figures.
From line 243 to 246, there were typographical error. The sentence of “The estimated increases in SPD and SPM intake in the intervention group were 22.00±9.56 and 96.63±47.70 µmol/day, respectively, while those in the control group decreased slightly by -0.34±1.48 and -2.63±11.60 µmol/day, respectively” in the first submission has been fixed as “The estimated increases in SPD and SPM intake in the intervention group were 96.63±47.70 and 22.00±9.56 µmol/day, respectively, while those in the control group decreased slightly by -2.63±11.60 and 0.34±1.48µmol/day, respectively”
In abstract, from line 22 to 23, there also were typographical error. We replaced the sentence of “The estimated increases in spermidine and spermine intakes were 22.00±9.56 and 96.63±47.70 µmol per day in the intervention group, while no changes were observed in the control group” by “The estimated increases in spermidine and spermine intakes were 96.63±47.70 and 22.00±9.56 µmol per day in the intervention group, while no changes were observed in the control group”
In addition, from line 404-405, there were typographical error. We replaced the sentence “foods in the Japanese population are 36 and 74 µmol/day, respectively” by “foods in the Japanese population are 74 and 36 µmol/day, respectively”
- The last two sentences of the abstract are highly speculative and overstated and should be omitted. The authors cannot talk about aging-associated pro-inflammatory status, when they did not include young controls.
The comments of the last part of the abstract was based on many recent studies on gene methylation and aging and our previous in vitro and in vivo studies. We changed latter half of the abstract.
- The authors should clarify for instance for Figure 2, if the correlations were calculated across both control and intervention groups. At least the number of data points suggests so. If this is the case, then – in my understanding – the correlations cannot be presented like this and should be evaluated only within the groups. Also, the groups should be indicated (by colour or symbol). Else, there is a high risk of creating spurious correlation artifacts. Hence, the correlations must be recalculated and additionally presented at least for the intervention group alone.
In control group, polyamine intake in several volunteers changed after intervention. Therefore we analyzed changes of polyamine intake and markers in all volunteers. The comparison of two groups were evaluated in other part.
- Why did the authors not measure standard inflammatory blood markers like IFNs, TNF-alpha and so on?
I cannot understand the word “standard inflammatory markers”. Hs-CRP is well established inflammatory marker. TNF and IFN were produced upon stimulation by pathogens, and increases in their blood concentrations are very short. We consider it is not appropriate to use such mediators.
- Is the exact nutritional composition of the natto used in the study known? If so, this should be mentioned here, as there are likely other bioactive compounds present.
Natto is made of soybeas, and the nutritional composition is well known. Basically, the nutritional composition is very similar to soybeans. We want to focus on the polyamine intake, therefore we consider it is confusing to discuss about all nutritional composition in natto.
- The suggested daily additional intake of spermidine via the natto seems very high (>12mg for 1 pack, >24mg for 2 packs). This is way beyond what other currently running clinical trials use as spermidine supplementation. Has there been any kind of safety evaluation for this amount of daily polyamine intake? This needs to be commented on, especially regarding the cancer relevance. The authors need to justify and comment on the dosage selection. Also, were there any adverse events recorded? Is this information available anywhere?
Yes, polyamine concentrations in newly developed natto contains a large amount of polyamine. Polyamine rich food contains a large amount of polyamines. For example, groundnut usually contains almost 400 nmol/gram and mushrooms (edible) contain 1,000nmol/gram of spermidine. The mean spermidine concentration of soybeans is about 800 nmol/g. We measured polyamine concentrations in many soybeans and we found a soybean of which polyamine concentrations (1,400 nmol/g SPD and 190 nmol/g SPM) are higher than other soybeans. The amount of polyamine from natto is not far from toxic. If you know subacute and acute toxicity of spermidine, you can realize that the amount is very safe. These are described in polyamine textbook. The amount of polyamine taken by volunteers is much lower (estimated about 1/5000 to 1/10000) than the amount of subacute and acute toxicities. Therefore, we consider it is not necessary to mention about toxicity.
- Did the authors perform power analyses before recruiting? Please include this information.
No, because there was no preceding experiment.
- Is there an estimation of how much spermidine and spermine the diet of the study cohort contained over the study and what percentage the natto-supplementation increased over baseline intake levels? Should be possible to estimate based on the food records. This would be a crucial information for such studies.
More than half of foods we usually consume contain a small amount of polyamine or no polyamine. And, many of the foods people, not only volunteers, usually consume do not change much. Therefore, we analyzed polyamine concentrations in foods which are different before and after intervention.
- Blood levels of polyamines apparently do not correlate well with the increased polyamine intake. Why do the authors continue correlating blood polyamine changes with other parameters? Why don’t they concentrate on the intake data?
We also compared polyamine intake and other parameters. However, no correlations were found.
Minor
- Please explain “reinforced” natto, what does it exactly mean technically?
We replace “reinforced” by “high”.
- The link to the trial registration does not exist/work.
I confirmed that the trial registration does exist. However, the site was a Japanese version. We replaced the URL by URL of English registration site.
- Title of registered study does not match with the one in the Materials and Methods section. Please comment on this and clarify.
We changed registration name after first submission to the JMA. And we described the name of trial of initial registration. We replaced title name as it is in the JMA-IIA00233. (Line 99 to 101)
- In the cancer field there is some concern of polyamines enhancing tumor growth. The authors should address this point in the discussion. This is particularly important as the dosage which was used is extremely high.
At the time of registration, volunteers were asked whether they are suffering cancer or other proliferative diseases. When they have such diseases, we cannot accept them to join the study. The dosage contained in natto used in the study is high, but not extremely high to put volunteers’ health at risk. As written in the manuscript, the protocol and document was approved by Ethics Committee of Jichi Medical University.
- Please include a cohort description table.
Cohort description can be found in the manuscript body. The study design is very simple and the number of participants was small, it is easy for readers to understand the study overview when referring the manuscript body and Table 1.
- Table 1: Are the indicated p-values the only significant ones?
Yes. All the data of the two groups were evaluated.
- Tables 1 and 2 basically show the same data, should be combined.
Yes, basically, table 1 and 2 show the same data. However, Table 2 provide information of changes in parameters after intervention. Therefore, Table 2 help readers to recognize the changes of parameters of both groups.
- There is a large discrepancy of literature on the age-dependency of polyamine concentration in murine tissue and human samples. The authors concentrate on literature supporting their own data but should also include other reports. Especially, the data should be discussed in view of Pekar et al. and Pucciarelli et al.
When you read papers in which polyamine levels were measured, majority of papers showed that there is no age-dependent decline in polyamine concentrations in blood, tissues, organs and urine after we grew up. In addition, spermidine concentrations in some organs increase with aging. These findings were already confirmed by many polyamine investigators. The age dependent decline in polyamine concentrations is only observed in the developing young stage. This article is not a review article, therefor it is simple to refer papers of which results are comparable to those of the present study. And important thing is that majority of the previous study indicate no age-dependent decline. One big problem is that when serum polyamine concentrations are measured, only a tiny peak of spermine on HPLC is noticed. Serum polyamine concentration is only about 1% of whole blood spermine concentration, because almost all polyamines are attached to cell components. Therefore, it is hard to measure accurate polyamine, especially spermine, concentrations in serum, and we think it is not good idea to compare concentrations. Tiny peak of polyamine, depicted as very tiny peak and sometime looks like the shaking of the base line, hardly provide accurate measurements. We do not ignore their studies, we just focused on the results of our studies and disused referring papers that support our obtained results.
- The discussion should include a dedicated and extended limitations part.
We already discussed the limitation of the study in the discussion section of the initial submission. (line 397 to 412)
- Are there records of study compliance? E.g. how often did participants in the intervention group miss the natto intake goal?
We described in the original manuscript.
- Line 48: “Nutritional components and macrobiotic…” A word seems to be missing after macrobiotic.
In the original manuscript, description is like that “Nutritional components and macrobiotic in healthy diets have biological roles in inhibiting an aging-associated pro-inflammatory status, ……..”
I do not think word is missing. Please indicate.
- Line 52: Please provide information about polyamine content estimations for both diets.
Please refer paper 7 in which content is created based on the measured data of previous studies done by several groups.
- Lines 71-72: Incomplete sentence, verb is missing.
Thank you. A word “by” was deleted. The SPM concentration-dependent decrease in LFA-1 expression accompanied a decrease in the adhesion of peripheral blood mononuclear cells (PBMCs)
- At the first mention of the inflammation parameters their relevance should be briefly introduced.
LFA-1 is one of the very important and established marker of pro-inflammatory
- Table legends should include explanations of the abbreviations.
We added information to the all table legends.
- I recommend indicating r and p-values in the correlation plots.
We do not like to put r and p values in the graph. Instead, we explained in the figure legends.
- Lines 213-214: Unit missing for hs-CRP.
From 117 to 118 in the revised ms, unit for hs-CRP was described as “High sensitivity C-reactive protein (hs-CRP) (ng/mL) was measured by …..”.
- Table 4: What is the unit of the numbers? How does this table relate to the data shown in Table 2?
In the table legends of the original manuscript, we described as “Numerical changes of values at each point from pre-intervention values were expressed as mean±SD.”
- Figure 3 seems to cherry-pick time points, why not systematically analyzing the correlations across all time points?
We analyzed the correlations of at time points. Figure 3 shows the positive data.
- Another point that needs to be discussed in a limitations section is that most of the significant observations have rather high p-values, while no corrections for multiple comparisons are applied. Thus, the conclusions should be considered very preliminary and with caution. Given this, it needs to be made very clear that this study was of explorative nature.
We analyzed the correlations of at time points.
- Discussion Line 352 should be referenced.
As described in the manuscript of first submission, it is already referenced [13]
- Line 352: Decide to use plural or singular.
“studies” in line 352 in the original manuscript was replaced by “study”
- The manuscript should be checked by a native English-speaking person.
We already asked a native English-speaker and submitted a manuscript after English check. We are very happy if you correct it.
- The molecular mechanisms of polyamines are wide and somewhat elusive, especially for external supplementation. The authors did not address some key molecular aspects of external polyamine supply, e.g. hypusination of eIF5A or autophagy induction, which could at least indirectly by examined in the isolated PBMCs. If time allows, this would increase the strength of the study but if it’s technically not possible, the variety of potential molecular mechanisms should at least be mentioned in the discussion. Also, if blood cells are available, the authors could determine polyamine levels in those samples, as they might respond differently than whole blood to the increased polyamine intake.
Thank you very much for your valuable comments.
- The authors should show unstained controls, at least examples for e.g. Fig 1 and so on.
We cannot understand the meaning of “unstained control”.
- Line 229-230: estimated natto polyamine levels should be referenced.
There was a legends for Table 2 in the first submission. We cannot understand which sentence should be referenced.
- The data on HDL/LDL should also be mentioned in the abstract.
We cannot understand your comment. No changes or effects on HDL/LDL was found. Why we should mention.
We changed latter half of abstract extensively to simplify.
Reviewer 2 Report
This study investigated whether increased polyamine intake elevated blood polyamine levels in humans and provoked the biological effects of polyamine. The present study has a scientific contribution to the related research field. However, there are some issues to be figured out before publication. My comments are as follows:
- The section of Abstract is not enough condensed. It should be simplified and re-organized.
- There are too much keywords. Generally, a total of 5 keywords should be fine. Besides, the abbreviation should not occur in the keywords.
- Line 84 and line 89, there is a repetition. Please specify the novelty and significance of this study in the last paragraph in the section of Introduction.
- Line 111, SRL and line 121 PBMCs, please give full name and some details about the isolation of PBMCs.
- The legends of all figures and tables should be an independent part. Please give some necessary details and add full name of some abbreviations.
- For each part of Results, authors should give a brief summary at the last paragraph.
- Please add the section of Conclusion.
Author Response
1. The section of Abstract is not enough condensed. It should be simplified and re-organized.
We changed latter half of abstract extensively to simplify it.
2. There are too much keywords. Generally, a total of 5 keywords should be fine. Besides, the abbreviation should not occur in the keywords.
We deleted 5 keywords and the abbreviations from original submission. (marked)
3. Line 84 and line 89, there is a repetition. Please specify the novelty and significance of this study in the last paragraph in the section of Introduction.
We deleted the last sentence of the introduction section. We added “in humans” to the end of the first sentence of the last paragraph, and added a sentence “To test the effects of high polyamine diet, natto was used for the study”.
4. Line 111, SRL and line 121 PBMCs, please give full name and some details about the isolation of PBMCs.
SRL is a company of clinical laboratory testing. We added “a clinical laboratory testing company” in ( ). PBMCs was defined (line 71 in revised ms.). We added sentences about the isolation of PBMCs. (marked yellow)
5. The legends of all figures and tables should be an independent part. Please give some necessary details and add full name of some abbreviations.
We added details and full name of abbreviations. (marked in yollow)
6. For each part of Results, authors should give a brief summary at the last paragraph.
This is the first time to receive such request, and there seems no request in the template of Medical Sciences. It is very difficult for us to a brief summery in each result section. Please allow me to skip this request.
7. Please add the section of Conclusion.
We added the section of conclusion.
Round 2
Reviewer 1 Report
New reviewer responses in bold.
- I think the title is misleading and I would suggest changing it to something like „Intake of natto with increased polyamine content elevates…”
Thank you very much. We change the title of our manuscript referring to your opinions. (marked by broad yellow line)
Reviewer Response: As already pointed out, I think that the title still is misleading in the sense, that the study examined the effects of increased, polyamine-rich natto intake. Currently, the title suggests that spermidine and spermine were directly increased in the diet. As natto contains a lot of other ingredients I would propose a more direct, precise title. If the authors are not willing to discuss this point, I leave it up to the editor to decide.
- The main conclusion of title and abstract (“The intervention increases spermine levels.”) is based on repeated t-tests and/or mann-whitney-u tests separately at each timepoint. I am not convinced that this is the best choice of analysis and would recommend consulting a dedicated statistician about this point.
We asked the doctor who is expertise in this field, and we examined t-test and Mann Whitney tests separately at each time point. No p-value indicated when there is no significance or difference.
- The manuscript lacks and completely ignores significant references to work by other groups with spermidine and polyamines in inflammation research, model organisms, aging research, autophagy induction and clinical trials. The authors must extend the inclusion of relevant work and lengthen their discussion to relevant preclinical and clinical studies, as well as recent reviews which discuss the field.
We do not ignore works by other groups. When we refer many previous papers in which spermidine concentrations in tissues, organs, blood, urine, and semen were measured, the effects of spermine (as you know, spermidine has little ability to suppress inflammation) on inflammation, and the experimental results of model organisms and aging research, there must be too many citations. This article is not a review article, therefore, we cited our review articles in which we referred many papers reported not only by us but by other investigators.
Aging-associated decrease in polyamine concentrations described in the title or the abstract only indicate that the decrease is observed only during developmental and growth period. When you read the body of many previous papers concerning on the polyamine concentrations in tissues, organs, blood, and urine, you can find that majority of the previous paper clearly show that spermidine does not change with aging in adult. Moreover, in some reports, it was shown that spermidine concentrations in prostate and semen in humans and in pancreas in animals increase (not increase) with aging. When you read previous papers, you realize that the majority of papers clearly showed no age-dependent decrease in spermidine concentration in adult. The one problem we concern is that when serum polyamine concentrations is measured by HPLC, it sometimes hard to detect peak of polyamine, especially spermine. In some case with lower polyamine concentrations, spermine peak is detected like a shaking of the base line of HPLC. We consider it is very difficult to determine accurate polyamine concentrations using such unclear peak. As you know, serum contains only 1% of blood polyamine. Majority of the polyamine is contained in blood cells. These facts indicate that when hemolysis, no matter how slight, occurred in the blood sample, serum levels of polyamine were affected significantly.
The other curious thing is that many food components, such as so-called anti-oxidant substances and anti-oxidant vitamins failed to extend life span of mammals in spite of the fact they are absorbed in the body and they significantly activate autophagy.
In the near future, I will publish a review article in which recent works on spermidine and polyamines in inflammation research, model organisms, aging research, autophagy induction and clinical trials.
Reviewer Response: Agree to disagree. The reviewers include a lot of self-citations in the paper, but ignore recent advances in the field. Whether the authors will publish reviews in the future is of no interest for the present manuscript.
For instance.: 10.7554/eLife.57950; 10.1146/annurev-nutr-120419-015419; 10.18632/aging.101354;
- The information on increased polyamine intake in the abstract does not match the information from the main text or figures. Please clarify which information is the right one. For instance, compare Line 233 with Figure 2d-i. Also if the lowest intake per day is 1 pack of natto (1880nmol/g, 45g package size), then the lower limit should be roughly 84.6µmol spermidine. Still, the average increase is stated as 22µmol/day and this also does not fit with the mentioned figures.
From line 243 to 246, there were typographical error. The sentence of “The estimated increases in SPD and SPM intake in the intervention group were 22.00±9.56 and 96.63±47.70 µmol/day, respectively, while those in the control group decreased slightly by -0.34±1.48 and -2.63±11.60 µmol/day, respectively” in the first submission has been fixed as “The estimated increases in SPD and SPM intake in the intervention group were 96.63±47.70 and 22.00±9.56 µmol/day, respectively, while those in the control group decreased slightly by -2.63±11.60 and 0.34±1.48µmol/day, respectively”
In abstract, from line 22 to 23, there also were typographical error. We replaced the sentence of “The estimated increases in spermidine and spermine intakes were 22.00±9.56 and 96.63±47.70 µmol per day in the intervention group, while no changes were observed in the control group” by “The estimated increases in spermidine and spermine intakes were 96.63±47.70 and 22.00±9.56 µmol per day in the intervention group, while no changes were observed in the control group”
In addition, from line 404-405, there were typographical error. We replaced the sentence “foods in the Japanese population are 36 and 74 µmol/day, respectively” by “foods in the Japanese population are 74 and 36 µmol/day, respectively”
- The last two sentences of the abstract are highly speculative and overstated and should be omitted. The authors cannot talk about aging-associated pro-inflammatory status, when they did not include young controls.
The comments of the last part of the abstract was based on many recent studies on gene methylation and aging and our previous in vitro and in vivo studies. We changed latter half of the abstract.
- The authors should clarify for instance for Figure 2, if the correlations were calculated across both control and intervention groups. At least the number of data points suggests so. If this is the case, then – in my understanding – the correlations cannot be presented like this and should be evaluated only within the groups. Also, the groups should be indicated (by colour or symbol). Else, there is a high risk of creating spurious correlation artifacts. Hence, the correlations must be recalculated and additionally presented at least for the intervention group alone.
In control group, polyamine intake in several volunteers changed after intervention. Therefore we analyzed changes of polyamine intake and markers in all volunteers. The comparison of two groups were evaluated in other part.
Reviewer Response: The authors cannot calculate the presented linear correlations across both groups without taking the existence subgroups into account. This is a common, but very misleading mistake. Only if the correlations hold true within the intervention group, this would be allowed. At least indicate the group allocation as suggested before, by different colors or symbols so that the statistically wise reader can interpret the data. The clustering of data points already suggests that the linear correlations are spurious ones, only existing because of mixing controls and intervention groups. This has been shown multiple times before. Please see, for example, https://elifesciences.org/articles/48175 Fig. 2E and 2F. The authors must discuss this in the manuscript. There is absolutely no reason why the authors would not indicate the group allocation in the figure.
- Why did the authors not measure standard inflammatory blood markers like IFNs, TNF-alpha and so on?
I cannot understand the word “standard inflammatory markers”. Hs-CRP is well established inflammatory marker. TNF and IFN were produced upon stimulation by pathogens, and increases in their blood concentrations are very short. We consider it is not appropriate to use such mediators.
- Is the exact nutritional composition of the natto used in the study known? If so, this should be mentioned here, as there are likely other bioactive compounds present.
Natto is made of soybeas, and the nutritional composition is well known. Basically, the nutritional composition is very similar to soybeans. We want to focus on the polyamine intake, therefore we consider it is confusing to discuss about all nutritional composition in natto.
- The suggested daily additional intake of spermidine via the natto seems very high (>12mg for 1 pack, >24mg for 2 packs). This is way beyond what other currently running clinical trials use as spermidine supplementation. Has there been any kind of safety evaluation for this amount of daily polyamine intake? This needs to be commented on, especially regarding the cancer relevance. The authors need to justify and comment on the dosage selection. Also, were there any adverse events recorded? Is this information available anywhere?
Yes, polyamine concentrations in newly developed natto contains a large amount of polyamine. Polyamine rich food contains a large amount of polyamines. For example, groundnut usually contains almost 400 nmol/gram and mushrooms (edible) contain 1,000nmol/gram of spermidine. The mean spermidine concentration of soybeans is about 800 nmol/g. We measured polyamine concentrations in many soybeans and we found a soybean of which polyamine concentrations (1,400 nmol/g SPD and 190 nmol/g SPM) are higher than other soybeans. The amount of polyamine from natto is not far from toxic. If you know subacute and acute toxicity of spermidine, you can realize that the amount is very safe. These are described in polyamine textbook. The amount of polyamine taken by volunteers is much lower (estimated about 1/5000 to 1/10000) than the amount of subacute and acute toxicities. Therefore, we consider it is not necessary to mention about toxicity.
- Did the authors perform power analyses before recruiting? Please include this information.
No, because there was no preceding experiment.
- Is there an estimation of how much spermidine and spermine the diet of the study cohort contained over the study and what percentage the natto-supplementation increased over baseline intake levels? Should be possible to estimate based on the food records. This would be a crucial information for such studies.
More than half of foods we usually consume contain a small amount of polyamine or no polyamine. And, many of the foods people, not only volunteers, usually consume do not change much. Therefore, we analyzed polyamine concentrations in foods which are different before and after intervention.
- Blood levels of polyamines apparently do not correlate well with the increased polyamine intake. Why do the authors continue correlating blood polyamine changes with other parameters? Why don’t they concentrate on the intake data?
We also compared polyamine intake and other parameters. However, no correlations were found.
Minor
- Please explain “reinforced” natto, what does it exactly mean technically?
We replace “reinforced” by “high”.
- The link to the trial registration does not exist/work.
I confirmed that the trial registration does exist. However, the site was a Japanese version. We replaced the URL by URL of English registration site.
Reviewer response: Works now, thanks.
- Title of registered study does not match with the one in the Materials and Methods section. Please comment on this and clarify.
We changed registration name after first submission to the JMA. And we described the name of trial of initial registration. We replaced title name as it is in the JMA-IIA00233. (Line 99 to 101)
- In the cancer field there is some concern of polyamines enhancing tumor growth. The authors should address this point in the discussion. This is particularly important as the dosage which was used is extremely high.
At the time of registration, volunteers were asked whether they are suffering cancer or other proliferative diseases. When they have such diseases, we cannot accept them to join the study. The dosage contained in natto used in the study is high, but not extremely high to put volunteers’ health at risk. As written in the manuscript, the protocol and document was approved by Ethics Committee of Jichi Medical University.
- Please include a cohort description table.
Cohort description can be found in the manuscript body. The study design is very simple and the number of participants was small, it is easy for readers to understand the study overview when referring the manuscript body and Table 1.
- Table 1: Are the indicated p-values the only significant ones?
Yes. All the data of the two groups were evaluated.
- Tables 1 and 2 basically show the same data, should be combined.
Yes, basically, table 1 and 2 show the same data. However, Table 2 provide information of changes in parameters after intervention. Therefore, Table 2 help readers to recognize the changes of parameters of both groups.
- There is a large discrepancy of literature on the age-dependency of polyamine concentration in murine tissue and human samples. The authors concentrate on literature supporting their own data but should also include other reports. Especially, the data should be discussed in view of Pekar et al. and Pucciarelli et al.
When you read papers in which polyamine levels were measured, majority of papers showed that there is no age-dependent decline in polyamine concentrations in blood, tissues, organs and urine after we grew up. In addition, spermidine concentrations in some organs increase with aging. These findings were already confirmed by many polyamine investigators. The age dependent decline in polyamine concentrations is only observed in the developing young stage. This article is not a review article, therefor it is simple to refer papers of which results are comparable to those of the present study. And important thing is that majority of the previous study indicate no age-dependent decline. One big problem is that when serum polyamine concentrations are measured, only a tiny peak of spermine on HPLC is noticed. Serum polyamine concentration is only about 1% of whole blood spermine concentration, because almost all polyamines are attached to cell components. Therefore, it is hard to measure accurate polyamine, especially spermine, concentrations in serum, and we think it is not good idea to compare concentrations. Tiny peak of polyamine, depicted as very tiny peak and sometime looks like the shaking of the base line, hardly provide accurate measurements. We do not ignore their studies, we just focused on the results of our studies and disused referring papers that support our obtained results.
- The discussion should include a dedicated and extended limitations part.
We already discussed the limitation of the study in the discussion section of the initial submission. (line 397 to 412)
- Are there records of study compliance? E.g. how often did participants in the intervention group miss the natto intake goal?
We described in the original manuscript.
- Line 48: “Nutritional components and macrobiotic…” A word seems to be missing after macrobiotic.
In the original manuscript, description is like that “Nutritional components and macrobiotic in healthy diets have biological roles in inhibiting an aging-associated pro-inflammatory status, ……..”
I do not think word is missing. Please indicate.
Reviewer response: „Macrobiotic“ is an adjective. You can either write “Nutritional and macrobiotic components…” or “Nutritional components and macrobiotics…”, where “macrobiotics” can be used in plural as a noun.
- Line 52: Please provide information about polyamine content estimations for both diets.
Please refer paper 7 in which content is created based on the measured data of previous studies done by several groups.
- Lines 71-72: Incomplete sentence, verb is missing.
Thank you. A word “by” was deleted. The SPM concentration-dependent decrease in LFA-1 expression accompanied a decrease in the adhesion of peripheral blood mononuclear cells (PBMCs)
- At the first mention of the inflammation parameters their relevance should be briefly introduced.
LFA-1 is one of the very important and established marker of pro-inflammatory
- Table legends should include explanations of the abbreviations.
We added information to the all table legends.
- I recommend indicating r and p-values in the correlation plots.
We do not like to put r and p values in the graph. Instead, we explained in the figure legends.
- Lines 213-214: Unit missing for hs-CRP.
From 117 to 118 in the revised ms, unit for hs-CRP was described as “High sensitivity C-reactive protein (hs-CRP) (ng/mL) was measured by …..”.
- Table 4: What is the unit of the numbers? How does this table relate to the data shown in Table 2?
In the table legends of the original manuscript, we described as “Numerical changes of values at each point from pre-intervention values were expressed as mean±SD.”
- Figure 3 seems to cherry-pick time points, why not systematically analyzing the correlations across all time points?
We analyzed the correlations of at time points. Figure 3 shows the positive data.
- Another point that needs to be discussed in a limitations section is that most of the significant observations have rather high p-values, while no corrections for multiple comparisons are applied. Thus, the conclusions should be considered very preliminary and with caution. Given this, it needs to be made very clear that this study was of explorative nature.
We analyzed the correlations of at time points.
- Discussion Line 352 should be referenced.
As described in the manuscript of first submission, it is already referenced [13]
- Line 352: Decide to use plural or singular.
“studies” in line 352 in the original manuscript was replaced by “study”
- The manuscript should be checked by a native English-speaking person.
We already asked a native English-speaker and submitted a manuscript after English check. We are very happy if you correct it.
- The molecular mechanisms of polyamines are wide and somewhat elusive, especially for external supplementation. The authors did not address some key molecular aspects of external polyamine supply, e.g. hypusination of eIF5A or autophagy induction, which could at least indirectly by examined in the isolated PBMCs. If time allows, this would increase the strength of the study but if it’s technically not possible, the variety of potential molecular mechanisms should at least be mentioned in the discussion. Also, if blood cells are available, the authors could determine polyamine levels in those samples, as they might respond differently than whole blood to the increased polyamine intake.
Thank you very much for your valuable comments.
- The authors should show unstained controls, at least examples for e.g. Fig 1 and so on.
We cannot understand the meaning of “unstained control”.
- Line 229-230: estimated natto polyamine levels should be referenced.
There was a legends for Table 2 in the first submission. We cannot understand which sentence should be referenced.
- The data on HDL/LDL should also be mentioned in the abstract.
We cannot understand your comment. No changes or effects on HDL/LDL was found. Why we should mention.
Reviewer Response: The authors included the following primary outcomes in their trial registration: „Changes in blood polyamine levels. Changes of blood data induced by polyamine, sucha as LFA-1(lymphocyte function associated antigen 1) expression on immune cells, blastoid transformation of lymphocyte by con-A, HDL, LDL, and CRP.“ Thus, including the information in the abstract should definitely be considered, regardless if there were significant differences or not.
We changed latter half of abstract extensively to simplify.
Additional comments
- Primary outcomes, as registered, should be clearly indicated in the manuscript. Either in the introduction, abstract or discussion. This is standard for clinical trials.
- Line 30: „The results in this study WERE similar…”

Author Response
Revised part is marked in broad red line.
Reviewer Response: As already pointed out, I think that the title still is misleading in the sense, that the study examined the effects of increased, polyamine-rich natto intake. Currently, the title suggests that spermidine and spermine were directly increased in the diet. As natto contains a lot of other ingredients I would propose a more direct, precise title. If the authors are not willing to discuss this point, I leave it up to the editor to decide.
Majority of readers who are interested in this field know that it is not allowed to add synthetic polyamine into food. Therefore, I do not think the title mislead the sense. Off course, all food contain “a lot of other ingredients”. Natto also contains many other ingredients. However, many volunteers in the study (even in the control group) eat natto. Our concern is the difference the amount of polyamine in diet including natto between the study and the control group. Besides, almost all readers except Japanese do not know about natto. Therefore, using unfamiliar word may provoke confusion for the reader.
Reviewer Response: Agree to disagree. The reviewers include a lot of self-citations in the paper, but ignore recent advances in the field. Whether the authors will publish reviews in the future is of no interest for the present manuscript.
We respond this issue in the previous revision. New publication or publish in journal with high impact factor is not the same thing as advances. Many previous papers provoked confusion and mislead advances, especially in the field of nutrition ingredients that were almost blindly believed to help extend human life span. We just referred our review articles in which many papers previously published by excellent scientists in polyamine field. And, the important thing is that the results of the majority of these previous papers support our results in the present study. In addition, the reason why we hesitate to refer recent publication was described in the previous response. Therefore, your opinion of “self-citations” is an off-the point view. Again, too many citations in the original paper should be avoided.
Reviewer Response: The authors cannot calculate the presented linear correlations across both groups without taking the existence subgroups into account. This is a common, but very misleading mistake. Only if the correlations hold true within the intervention group, this would be allowed. At least indicate the group allocation as suggested before, by different colors or symbols so that the statistically wise reader can interpret the data. The clustering of data points already suggests that the linear correlations are spurious ones, only existing because of mixing controls and intervention groups. This has been shown multiple times before. Please see, for example, https://elifesciences.org/articles/48175 Fig. 2E and 2F. The authors must discuss this in the manuscript. There is absolutely no reason why the authors would not indicate the group allocation in the figure.
The question is exactly the same as the first requirement of revision. Again, not only volunteers in the study group but many volunteers in the control group eat commercially available natto. It is important to consider the relationship between the changes in the amount of polyamine intake and the changes in blood polyamine levels. Therefore, we tested among all volunteers. We already responded the issue.
Reviewer response: „Macrobiotic“ is an adjective. You can either write “Nutritional and macrobiotic components…” or “Nutritional components and macrobiotics…”, where “macrobiotics” can be used in plural as a noun.
Thank you very much. I appreciate your English editing.
Reviewer Response: The authors included the following primary outcomes in their trial registration: „Changes in blood polyamine levels. Changes of blood data induced by polyamine, sucha as LFA-1(lymphocyte function associated antigen 1) expression on immune cells, blastoid transformation of lymphocyte by con-A, HDL, LDL, and CRP.“ Thus, including the information in the abstract should definitely be considered, regardless if there were significant differences or not.
Why did not point out the issue in the initial review process. You just mentioned about the ratio of LDL and HDL. However, in the initial submission, we stated the primary out comes in the end of introduction “This study examines whether increased polyamine intake elevates blood polyamine levels in humans and provokes the biological effects of polyamine in humans.” And, in the majority part of the introduction, we mentioned about biological effects of polyamine, namely LFA-1 expression.
Additional comments
- Primary outcomes, as registered, should be clearly indicated in the manuscript. Either in the introduction, abstract or discussion. This is standard for clinical trials.
We already described the primary out comes in the end of introduction “This study examines whether increased polyamine intake elevates blood polyamine levels in humans and provokes the biological effects of polyamine in humans.” And, in the majority part of the introduction, we mentioned about biological effects of polyamine, namely LFA-1 expression.
- Line 30: „The results in this study WERE similar…”
Thank you very much. I appreciate your English editing.
Reviewer 2 Report
Thanks for your revised manuscript. I have no further comments.
Author Response
Thank you very much for your kind review.